# Temporal Comparative Transcriptome Analysis on Wheat Response to Acute Cd Toxicity at the Seedling Stage

**DOI:** 10.3390/plants12030642

**Published:** 2023-02-01

**Authors:** Imdad Ullah Zaid, Mohammad Faheem, Muhammad Amir Zia, Zaheer Abbas, Sabahat Noor, Ghulam Muhammad Ali, Zeeshan Haider

**Affiliations:** 1National Institute for Genomics and Advanced Biotechnology (NIGAB), National Agricultural Research Centre, Islamabad 45500, Pakistan; 2Nuclear Institute of Agriculture, Tando Jam 70050, Pakistan; 3Hebei Key Laboratory of Soil Ecology, Centre for Agricultural Resources Research, Institute of Genetics and Developmental Biology, Chinese Academy of Sciences, Shijiazhuang 050021, China

**Keywords:** seedling stage, wheat, Cd stress breeding, transcriptomic analysis, DEGs

## Abstract

Cadmium (Cd) is a non-essential and toxic metal that accumulates in plant’s tissues and diminishes plant growth and productivity. In the present study, differential root transcriptomic analysis was carried out to identify Cd stress-responsive gene networks and functional annotation under Cd stress in wheat seedlings. For this purpose, the Yannong 0428 wheat cultivar was incubated with 40 µm/L of CdCl_2_·2.5H_2_O for 6 h at three different seedling growth days. After the quality screening, using the Illumina Hiseq 2000 platform, more than 2482 million clean reads were retrieved. Following this, 84.8% to 89.3% of the clean reads at three time points under normal conditions and 86.5% to 89.1% of the reads from the Cd stress condition were mapped onto the wheat reference genome. In contrast, at three separate seedling growth days, the data analysis revealed a total of 6221 differentially expressed genes (DEGs), including 1543 (24.8%) up-regulated genes and 4678 (75.8%) down-regulated genes. In total, 120 DEGs were co-expressed throughout all the growth days, whereas 1096, 1088, and 2265 DEGs were found to be selectively up-/down-regulated at 7d, 14d, and 30d, respectively. However, the clustering of DEGs, through utilizing the Kyoto Encyclopedia of Genes and Genomes (KEGG), revealed that the DEGs in the metabolic category were frequently annotated for phenylpropanoid biosynthesis. In comparison, a considerable number of DEGs were linked to protein processing in the endoplasmic reticulum under the process of genetic information processing. Similarly, in categories in organismal systems and cellular processes, DEGs were found in plant hormone signal transduction pathways, and DEGs were identified in the plant–pathogen interaction pathway, respectively. However, DEGs in “endocytosis pathways” were enriched in environmental information processing. In addition, in-depth annotations of roughly specific heavy metal stress-response genes and pathways were also mined, and the expression patterns of eight DEGs were studied using quantitative real-time PCR. The results were congruent with the findings of RNA sequencing regarding transcript abundance in the studied wheat cultivar.

## 1. Introduction

Nearly one fifth of cultivated land in China is polluted by heavy metals [1,2]. The last 20 years have witnessed a remarkable rise in the heavy metal pollution of Chinese arable land through irrigation water that contains heavy metals, phosphate fertilizers, sludge and various agrochemicals. Heavy metals, particularly cadmium, mercury, lead and arsenic, may become toxic for plant growth and development if their concentrations outstrip the critical levels [3]. Among heavy metals, cadmium (Cd) is considered as a non-essential and highly toxic element for wheat metabolism, even at very low concentrations. Cd in the soil can easily be absorbed by wheat roots and transported to the shoot, leaf and grains, influencing the normal uptake of mineral nutrients [4,5]. It causes a wide range of deleterious effects on wheat transcriptomes and proteomes, resulting in oxidative stress, protein inactivation and reductions in plant biomass and, ultimately, grain yield [5,6,7,8]. Considering the importance of wheat for food and nutritional security, extensive efforts have been made toward developing wheat cultivars with high Cd tolerance.

Wheat consumption is considered to be one of the major sources of Cd intake for human beings. Recently, an increasing number of studies confirmed the presence of significant variations among wheat cultivars in response to different Cd stress concentrations [9,10,11,12,13]. Furthermore, the morphological, physiological and biochemical mechanisms of the Cd effect on wheat plants have been extensively studied [14,15,16]. The phenotypic and genotypic differences in wheat for Cd tolerance indicate the probability of the development of low-Cd wheat cultivars. However, the complex genome of wheat for controlling Cd tolerance is still not completely clear. Although, breeding low-Cd wheat cultivars by phenotyping for Cd tolerance in wheat is expensive and time-consuming [17]. The use of modern genomic tools for tagging the essential genes and molecular mechanisms of Cd in wheat could be a more sustainable approach for developing cultivars with low Cd levels.

DNA- and RNA-based next-generation sequencing (NGS) techniques have provided a quick and better understanding of the biological and molecular functions related to heavy metal stresses. With the ease of availability of sequencing platforms (RNA-Seq), expression profiling has been widely utilized to discover key differentially expressed genes (DEGs) involved in causing the heavy metal stresses. The identified DEGs can be annotated and used for further research using a reference genome sequence. RNA-Seq is a powerful and cost-effective molecular technology for underlying novel genes and their expression patterns in crop species that respond to Cd stress. Increasing studies that use transcriptome profiling have elucidated the Cd stress response in rice [18,19], cotton [20], soybean [21,22], maize [23,24], sorghum [25], and barley [26]. So far, quite a few studies have been carried out for wheat, revealing DEG responses to Cd stress and Cd-related signaling pathways. However, by conducting a genome-wide root transcriptome analysis of a low-Cd-accumulating wheat cultivar, Xiao et al. (2019) reported 1427 DEGs [27]. Yue et al. (2017) confirmed 8584 DEGs in wheat after 12 h of 100 μM high Cd stress [23]. Gene ontology enrichment analysis of these DEGs was mainly used to analyze stress tolerance and detoxification mechanisms, including metal transporters and biosynthesis-related hormones. Moreover, Zhou et al. (2019) also detected 76 novel expressed microRNAs and 1858 Cd-induced DEGs, after exposing two wheat cultivars to 100 μM of CdCl_2_ for 24 h [28].

We believe that the transcriptome dynamics in the wheat genome are still not well understood, and in particular, little is known about the Cd stress regulatory network at the seedling stage. Therefore, to gain a clear and detailed view of the transcriptomic response in wheat under Cd stress, we used next-generation transcriptome sequencing to profile the responses of Cd-tolerant wheat cultivars under 40 μM Cd stress. In the present study, a temporal comparative transcriptome analysis was carried out on the roots of a Cd-tolerant wheat cultivar, which differed at three seedling growth days under Cd-free and Cd-treated conditions. The main aims were to reveal the DEGs in the roots of Yannong 0428 in response to 6 h of Cd exposure, along with identifying the critical genes that are directly and indirectly associated with regulating wheat’s early responses to Cd stress. The results presented here will help to screen candidate genes of Cd stress in wheat at the seedling stage and for breeding low-Cd wheat cultivars.

## 2. Results

### 2.1. Primary Root Length at Different Time Points

The winter wheat cultivar Yannong 0428 is a Cd-tolerant cultivar, originating from North China, Shandong province. In our previous study, seedlings of Yannong 0428 revealed resistance against 40 ul/L Cd stress for root length, shoot length, root dry weight, shot dry weight and plant weight. The replicated mean root length of the seedlings at 7 days old was 14.2 cm, while at 14 days old, it was 29.8 cm and at 30 days old, it was 43.6 cm, respectively (Figure 1).

### 2.2. Illumina RNA Sequencing and Assembly Analysis

The RNA-Seq yielded a total of 2,482,825,626 (371 GB) cleaned raw reads, among which 1,269,754,350 (GB) reads were related to the normal conditions (nine libraries), while 1,213,071,276 reads were obtained from all the Cd stress libraries. Based on the average of three biological repeats, the libraries under normal conditions yielded 137.6 (7d) million reads, 145.7 (14d) and 139.90 (30d) million reads, while 124.8 (7d) million reads, 146.0 (14d) million reads and 133.6 (30d) million reads were obtained under Cd stress conditions. After filtering, the Q30 value of all the libraries ranged from 92.8% to 93.3%, which indicated the high quality of the sequence data (Table 1). Similarly, the GC contents of all the samples that represent both treatments were almost identical, with an overall average of 55.6%, depicting the usefulness of data for further bioinformatics analysis. Of all the clean reads obtained from the samples at different time points under normal conditions, 84.8% to 89.3% were mapped to the wheat reference genome, while 86.5% to 89.1% of reads from the Cd stress condition were mapped. The alignment analysis also showed that the uniquely mapped reads in normal and Cd stress samples ranged from 80.5% to 85.0% and 81.7% to 84.7%, respectively, whereas 4.2% of the remaining reads from each treatment were mapped to multiple regions (Table 1).

### 2.3. Comparative Transcriptome Analysis

The differential expression levels of each transcript in all the libraries were calculated based on the reading count values, and subsequently expressed as FPKM. High correlation coefficient values among the treatments (normal and Cd stress) at different seedling days represented the reproducibility and reliability of the sequence data (Figure 2). Subsequently, the comparative transcriptomic analysis of normal vs. Cd stress at three corresponding growth periods, using the criteria of FPKM > 1 of gene expression, revealed a total of 6221 differentially expressed genes (DEGs), among which 1644, 1704 and 2873 DEGs were found at the 7d, 14d and 30d growth stages, respectively (Table 2). These DEGs in each comparative group were further divided into up-regulated and down-regulated groups. The comparative DEG analysis identified a total of 1644 DEGs (462 up-regulated and 1182 down-regulated) at 7d, 1704 DEGs (245 up-regulated and 1459 down-regulated) at 14d and 2873 DEGs (514 up-regulated and 1559 down-regulated) at 30d. Concerning the comparison among stages, 1096, 1088, and 2265 DEGs were uniquely expressed, while 120 DEGs were commonly expressed at three different stages (7d, 14d and 30d). In contrast, the number of overlapping DEGs between other comparisons is also illustrated in the Venn diagram (Figure 3).

### 2.4. Functional Categorization of DEGs

#### 2.4.1. GO Enrichment Analysis

For functional characterization of the DEGs from each library, eight public databases were utilized, which annotated 1615 (98.24%), 1680 (98.59%), and 2816 (98.02%) DEGs at 7d, 14d and 30d, respectively, in at least one of these databases (Table 3). The GO functional enrichment analysis assigned 1295, 1442 and 2231 DEGs at 7d, 14d and 30d, respectively, to at least one GO term. Detailed analysis of GO annotation revealed that at 7d, 14d, and 30d, 51 GO terms (biological process, 21; cellular component, 16; molecular function, 14) were assigned, among which under the biological process category, the most enriched terms were related to metabolic, cellular and single-organism processes, response to stimulus and detoxification. Similarly, the cell, cell part, membrane, and organelle terms were enriched under the cellular component category. In contrast, each comparative group was highly enhanced by catalytic activity-, binding- and antioxidant activity-associated terms under the molecular function category (Figure 4).

#### 2.4.2. COG Enrichment Analysis

Based on the COG enrichment analysis, 714 (43.43%), 848 (49.77%) and 1106 (38.50%) DEGs in the 7d, 14d and 30d comparative groups were assigned to different functional classes of the COG database (Table 3). Among the 25 COG categories, the highly enriched classes at different time points (7d, 14d and 30d) of Cd stress included “carbohydrate transport and metabolism” post-translation modification, protein turnover, chaperons”; “secondary metabolites biosynthesis, transport and catabolism”; general function prediction only”; signal transduction mechanism” and “defense mechanisms” (Figure 5). It is noteworthy that among these commonly enriched COG categories, the class “carbohydrate transport and metabolism”, with 86, 121 and 154 DEGs, ranked top at the 7d, 14d and 30d growth stages, respectively. Two classes, including “energy production and conversion” and “amino acid transport and metabolism”, were significantly enriched at later stages (7d and 30d), as compared to the early stages of stress. On the other hand, the lowest number of DEGs was found to be associated with “replication, recombination and repair”; “cell motility”; and “intracellular trafficking, secretion and vesicular transport” COG classes during the course of Cd stress (Figure 5).

#### 2.4.3. KEGG Enrichment Analysis

To better understand the pathways associated with the Cd stress in wheat, a KEGG pathway analysis was conducted to assign pathways to different DEGs. The KEGG pathway annotation results revealed that 554 (33.70%), 658 (38.62%) and 891 (31.01%) DEGs could be assigned to different pathways following 7d, 14d and 30d of Cd stress, respectively (Table 3). At the initial stage of Cd stress, 82 (21.75%) DEGs in the metabolic category were associated with the “phenylpropanoid biosynthesis” pathway, while under the genetic information process category, a significant number of DEGs (79, i.e., 19.36%) were related to “protein processing in endoplasmic reticulum”. Similarly, under environmental information processing, organismal systems and cellular processes categories, 18 (4.77%) DEGs in the “plant hormone signal transduction pathway”, 19 DEGs (5.04%) in the “plant-pathogen interaction pathway”, and 10 DGEs (2.65%) in the “endocytosis pathways” were enriched, respectively (Figure 6). At the later stage (14d) of Cd stress, the KEGG pathway enrichment analysis assigned DEGs to new metabolic pathways, as compared to the initial stage, which included “carbon metabolism” (91 DGEs 20.00%), “biosynthesis of amino acids” (70 DGEs 15.38%), “gluconeogenesis” (70 DGEs 15.38%), and “carbon fixation” (36 DGEs 7.91%), in addition to “phenylpropanoid biosynthesis” (Figure 5). The ame KEGG annotation results were also obtained for 30d of Cd stress treatment, for which the number of DEGs in different metabolic pathways, including carbon metabolism” (52 DGEs 9.32%) and “biosynthesis of amino acids” (56 DGEs 10.04%), decreased. In contrast, the highest number of DEGs was found to be associated with the “phenylpropanoid biosynthesis” pathway, as compared to the 14d stress treatment (Figure 5B,C). Additionally, the highly enriched pathways in other categories were the same, with a greater number of DEGs for both the treatments, as was found at the initial Cd stress stages (Figure 6).

#### 2.4.4. KOG Enrichment Analysis

The frequency of DEGs at different time points of Cd stress in the respective KOG class is presented in Figure 6. The KOG annotation analysis assigned 786 (47.81%), 903 (52.99%) and 1291 (44.94%) DEGs obtained at 7d, 14d and 30d of Cd stress to different KOG categories. Among these KOG categories, the top category significantly enriched the following classes during Cd stress: “energy production and conversion”, “amino acid transport and metabolism”, “carbohydrate transport and metabolism”, “posttranslational modification, protein turnover, chaperons”, “general function prediction only”, and “signal transduction mechanism”. The top two KOG categories in terms of DEG frequency were “general function prediction only” and “posttranslational modification, protein turnover, chaperons”, with 144, 149, 235, 177, 110 and 220 DEGs at 7d, 14d and 30d of Cd stress, respectively (Figure 7). It was also observed that more DEGs were enriched for the “amino acid transport”, “inorganic ion transport and metabolism” and “signal transduction mechanism” categories at the later stages (14d and 30d), compared to the initial stages of Cd stress.

#### 2.4.5. eggNOG Enrichment Analysis

The eggNOG database was searched to assign the DEGs obtained at different stages of Cd stress to nested groups of orthologous genes for functional annotation. The blast search in the eggNOG database revealed that 1286 (78.22%), 1615 (94.78%) and 2641 (91.92%) DEGs from 7d, 14d and 30d Cd stress stages were associated with 26 eggNOG terms (Table 3). Among these transcripts, most of the DEGs, irrespective of the stress stage, could not be functionally annotated, as these belonged to the “function unknown” eggNOG class. The significantly enriched eggNOG terms across all the stress conditions were “carbohydrate transport and metabolism”, “transcription”; “posttranslational modification, protein turnover and chaperones” (Figure 8). Moreover, the DEGs related to “transcription”, “carbohydrate transport and metabolism”, and “signal transduction mechanism” were significantly enriched as the exposure time of Cd stress progressed (14d and 30d) (Figure 8).

### 2.5. DEGs in Key Pathway Associated with Cd

We further explored the transcriptomic and annotation data to investigate the specific responsive genes and pathways enriched upon Cd exposure for the Yannong 0428 wheat cultivar. The results of the in silico analysis identified a few particular transcripts associated with temporal Cd stress response.

#### 2.5.1. ABA Hormone-Related Genes

ABA hormones are usually considered to be signaling compounds that reveal substantial Cd stress tolerance accumulation by inducing several genes. In this study, 55 differently expressed ABA signaling genes of the abscisic acid receptor PYR/PYL family were annotated, among which 4 DEGs were up-regulated, while 51 depicted down-regulation. Detailed analysis of these DEGs revealed that 4 transcripts were expressed in seedlings at 7 days, 26 at 14 days and 22 at 30 days (Appendix A). These specific overrepresented genes were primarily involved in ABA hormone synthesis. Furthermore, four commonly expressed down-regulated genes (*TraesCS6B02G298500*, *TraesCS5D02G218300*, *TraesCS6A02G375800* and *TraesCS1A02G357200*) between three growth stages were highly enriched regarding secondary metabolism, envelope and carbohydrate transport and metabolism.

#### 2.5.2. Specific Heavy Metal Genes

Overall, the transcriptomic analysis of all the treatments identified 37 transcripts that were explicitly associated with heavy metals (Appendix A). These genes encoded heavy metal-associated prenylated plant proteins with different expression patterns; 8 were up-regulated, while the remaining 29 were down-regulated. The expression of these genes was found to be at its maximum (21 genes) in 30-day-old seedlings, followed by 14-day-old seedlings (11 genes) and 7-day-old seedlings (11 genes). The pathway enrichment analysis of heavy metal-related genes suggested that several modulated genes contribute to metal ion binding, transport, and transition metal ion homeostasis.

#### 2.5.3. Metal Ion Transport-Related Genes

Plant metal transporters play a unique role in Cd uptake, transport, and sequestration to prevent damage from non-essential metals. In this study, 50 transcripts for metal ion transport activity were identified, and were subsequently subjected to functional annotation (Appendix A). Most of the transcripts were down-regulated, suggesting that metal ion genes are affected by Cd stress. It was noteworthy that the expression of the genes was reported mainly for 30-day-old seedlings (28 genes), followed by 14-day-old seedlings (19 genes) and 7-day-old seedlings (6 genes). The responsive transcripts of the metal ion genes were primarily enriched with regard to metal ion transport activity, transition metal ion binding and inorganic ion transport and metabolism.

#### 2.5.4. Auxin-Related Genes

In this study, 60 auxin-related DEGs were identified in Yannong 0428 at different time points of Cd treatment. These auxin-associated genes showed different expression patterns, i.e., 7 were found to be up-regulated, while 53 were down-regulated. The data suggested that 21 DEGs were significantly expressed on the 7th day, whereas 46 DEGs were expressed at the 30-day-old seedling stage (Appendix A). However, none of the auxin-related genes were expressed in 14-day-old seedlings. Within both stages, eight DEGs, as a sign of the down-regulation of the AUX/IAA family, were common for auxin and auxin catabolic processes.

#### 2.5.5. ABC Transporter Pathway Genes

Our sequencing results detected 27 DEGs with ABC transport activity, with 14 up-regulated genes and 13 down-regulated genes. We found 12 genes in 7-day-old seedlings, 9 genes in 14-day-old seedlings, and 7 genes in 30-day-old seedlings (Appendix A). Most of the ABC transport genes were associated with ABC-2 family transporter proteins, contributing to transport and metabolism activity (ATP binding, lipoid and ABC transport).

#### 2.5.6. Peroxidase Activity-Related Genes

Our sequencing data detected 175 DEGs involved in peroxidase activity. Most of them were down-regulated following Cd exposure, revealing decreased POD activity in the roots. In the roots of wheat exposed to Cd at 30 days old, we observed a maximum number (85 genes) of DEGs, followed by 7-day-old seedlings (59 genes) and 77 DEGs for 14-day-old seedlings (Appendix A). Furthermore, two down-regulated genes (*TraesCS1D02G354100* and *TraesCS7D02G195100*) were shared among all three stress stages related to peroxidase synthesis.

### 2.6. Validation of Transcript Abundance

The expression profile of eight randomly selected genes was measured by qRT-PCR, using gene-specific primers to validate the transcriptomic results of Cd stress in wheat at different time points. Among these selected genes, six were up-regulated based on the transcriptomic results, which belonged to the “gene families’ response to auxin”; “ATP-binding”; “ADP-binding”; “transketolase activity”; and “serine-type endopeptidase inhibitor activity”. qRT-PCR also showed that the relative expression of these genes was significantly up-regulated (Figure 8). Similarly, the transcriptomic results indicated that two selected genes related to “peptidase activity” and cell membrane biogenesis were down-regulated. The same gene expression trend (down-regulation), as depicted by transcriptomic analysis, was demonstrated through qRT-PCR for these genes (Figure 9). Overall, the expression profiling of the randomly selected transcripts confirmed the authenticity, reliability, and reproducibility of the transcriptomic analysis.

## 3. Discussion

The primary goal of the present study was to figure out the crucial genes and complicated pathways involved in the response to Cd stress in the Yannong 0428 wheat cultivar at the three different seedling growth days. At the seedling stage, Cd stress is significant, since most crops are susceptible to stress. The comprehensive effects of Cd stress on wheat growth and development have been reviewed in our previous studies [8,29]. However, sufficient molecular knowledge is required to understand the complexity of Cd stress in crop plants. Functional genomics, proteomics, and physiological studies have made significant advances in this area in the last several decades. RNA-Seq-based next-generation sequencing (NGS) technology is an outstanding innovation, providing a comparatively economic and efficient way of mining crucial genes and their transcript level variations, along with their regulating factors with high accuracy.

Cd, as an abiotic stress factor, can alter gene expression in wheat at the seedling stage. The DEGs are considered to have a substantial effect on the acute potential transcription factors and tolerance genes related to metal transport and signal transduction [30]. The results indicated a total of 6221 DEGs found in all different sets after 6 h of Cd stress, with 26.4% expressed at 7 days, 27.3% at 14 days, and 46.1% at 30 days. While 24.5% of the DEGs were up-regulated, 76.1% were down-regulated, revealing that the wheat plant’s root gene expression pattern was altered due to Cd exposure. In addition, DEGs have been linked to stress-responsive gene families, such as Cd-responsive proteins, auxin-responsive proteins, auxin-induced proteins in root cultures, universal stress proteins, heavy metal-associated isoprenylated plant proteins, ABC transporter G family members, ROS production and signaling, HSP40- and HSP70-like proteins, ATP binding proteins, ATP citrate synthase, vacuolar ATP synthase subunit b, and vacuolar ATP synthase. Similarly, previous research has discovered that the expression of metal ion transporter genes correlates with Cd exposure [27]. The RNA-Seq approach might uncover novel Cd-responsive transporters by analyzing gene expression under different Cd exposures [31,32].

Out of all the constructed DEGs, 79% of the unigenes were categorized into 51 functional groupings, with 41.1% belonging to biological process, 31.3% belonging to molecular function, and 27.4% belonging to cellular components. Transcripts involved in the fundamental biological, cellular, and molecular approaches, such as signaling response (7.8%), metabolic activity (78.1%), and biological regulation (15%), were also discovered. This was consistent with previous studies of DEG(s) under Cd stress settings, and these findings imply that *T. aestivum* may contain several distinct genes that stimulate the response to Cd stress [27].

The KEGG database pathway annotation results revealed a total of 2103 DEGs, which were projected to map onto 5 major and 115 sub-metabolic pathways. In addition, “ABC transporters”, “plant hormone signal transduction”, “fatty acid degradation”, “biosynthesis of unsaturated fatty acid”, and “biosynthesis of amino acid” were among the stress signaling pathways studied. The genes involved in these activities are also crucial in cell metabolism [33,34,35]. It should also be noted that unsaturated fatty acids are drawn to various stimuli, controlling general defense mechanisms against biotic and abiotic threats [36]. However, in our study, 9 DEGs of fatty acid metabolism, 7 DEGs of fatty acid biosynthesis, 20 DEGs of fatty acid degradation, and 6 DEGs of unsaturated fatty acid biosynthesis were identified as promising candidates for future functional investigation. The six DEGs associated with unsaturated fatty acids elicited much attention in modulating anti-Cd stress functions and other stress processes [37]. Additionally, many previous studies have shown that ABC and abscisic acid (ABA) is involved in metal transport, sequestration, soil/hydroponic solution absorption, and efflux pumping at the plasma membrane [38,39]. Significant attention has been paid to the ways in which these transporters transport and regulate Cd tolerance in plants [40]. As a result, the current study discovered 14 DEGs of biological activities involved in encoding ABC proteins that were up-regulated and 13 down-regulated DEGs. We also found that when wheat was subjected to Cd stress, the expression of ABC transporter genes was also up-regulated [41]. At the same time, five ABC transporters (ABC transporter A, ABC transporter B, ABC transporter C, ABC transporter G, and ABC transporter I) were induced after Cd stress. As a comparison, ABC transporter genes were expressed at lower levels, although the ABC-2 transporter gene (TraesCS1B02G356100) was up-regulated in the G2 growth days. Similarly, the auxin-activated signaling pathway (GO: 0009734), auxin efflux (GO: 0010315), polar auxin transport (GO: 0009926), auxin response (GO: 0009733), auxin-mediated signaling pathway involved in phyllotactic patterning (GO: 0060774), auxin catabolic process (GO: 0009852), and auxin homeostasis were among the 61 down-regulated auxin-related DEGs (GO: 0010252). Our findings are consistent with the conclusions reported for maize, which showed that auxin pathway genes were predominantly down-regulated following Cd stress treatment [23]. On the other hand, the previous study identified that barley root tips were likewise damaged by short-term Cd treatment [42].

## 4. Material and Methods

### 4.1. Plant Material and Cd Stress Treatment

The wheat cultivar Yannong 0428, which originated in the Shandong province in North China, was used in this study. Healthy and uniform seeds of Yannong 0428 were selected and sterilized with 30% hydrogen peroxide (H_2_O_2_) solution for 10 min. After sterilization, the seeds were put on filter paper and placed in a petri dish to germinate in vermiculite for 48 h. Two-day-old seedlings were transferred into the holes of specialized germination bags, with hydroponic solution as the growth medium. After 7, 14 and 30 days of growth, half of the wheat seedlings were transferred to a nutrient solution with 40 µm/L of cadmium chloride monohydrate (CdCl_2_·2.5H_2_O). The seedlings of each time point were continuously treated with Cd solution for 6 h. The remaining seedlings at the same point were transferred to a nutrient solution without Cd as a control. At each time point of Cd treatment and the control, all the root tissues from the seedlings were collected in three biological replicates, immediately frozen in liquid nitrogen, and stored at −80 °C for RNA isolation and library construction. Furthermore, replicated root lengths of the seedlings were also measured.

### 4.2. RNA Isolation and Transcriptome Sequencing

Total RNA was extracted from 18 samples (7 days old seedlings Cd-treated and control, 14 days old Cd-treated and control and 30 days old seedlings Cd-treated and control × three biological replicates) using a NucleoSpin RNA Plant kit (MACHEREY-NAGEL GmbH & Co. KG, Düren, Germany), following the manufacturer’s protocol. RNA degradation and contamination were monitored on 1% agarose gel. RNA purity and concentration were measured using the NanoPhotometer^®^ and spectrophotometer (IMPLEN, CA, USA) and Qubit^®^ RNA Assay Kit in Qubit^®^2.0 Fluorometer (Life Technologies, CA, USA). Further, RNA integrity was also checked with the RNA Nano 6000 Assay Kit of the Agilent Bioanalyzer 2100 system (Agilent Technologies, CA, USA). For cDNA library construction and deep sequencing, sequencing libraries were proceeded with NEB Next^®^ Ultra™ RNA Library Prep Kit for Illumina^®^ (NEB, Ipswich, MA, USA) according to the manufacturer’s protocol and index codes were added to attribute sequences to each sample. The constructed libraries were sequenced on an Illumina Hiseq 2000 platform, generating paired-end reads.

### 4.3. Quality Check, Filtering and Alignment of RNA Sequence Data

The raw data for each sequencing sample obtained from the Illumina high-throughput platform were measured at both ends of the reads. The paired-end reads were cleaned by removing the reads that contained adapter sequences, poly-N, and low-quality reads. The reads’ quality was accessed with the FASTQC toolkit [43]. Residual adaptor sequences were removed from raw reads using cutadapt version 1.4.2 [44]. Simultaneously, the cleaned data’s base quality value (≥Q30%) and GC content were estimated. After removing the low-quality bases and filtering the reads, clean reads were aligned in parallel on the wheat reference genome sequences released by the International Wheat Genome Sequencing Consortium (IWGSC), using the HISAT2 software package [45] with default parameters. HISAT2 builds an index for genome sequences with Burrows–Wheeler transformation and the Ferragina–Manzini (FM) index that increases alignment speed and accuracy. After alignment, StringTie [46] was used to assemble the transcripts with mapped reads. All the 18 raw sequences were submitted to the National Genomics Data Center with PRJCA009646 as the accession number.

### 4.4. Identification and Cluster Analysis of Differentially Expressed Genes (DEGs)

Differential expression analysis of two conditions/groups was performed using the DESeq R package (1.10.1) [47]. DESeq provides statistical routines for determining differential expression in digital gene expression data, using a model based on the negative binomial distribution. The resulting *p* values were adjusted using Benjamini and Hochberg’s approach for controlling the false discovery rate. Genes with an adjusted *p*-value < 0.05 found by DESeq were assigned as differentially expressed genes (DEGs). DEGs were filtered to for hierarchical clustering analysis, so that the same or similar expression genes could be clustered together.

### 4.5. GO and KEGG Enrichment Analysis

Gene Ontology (GO) enrichment analysis of the DEGs was implemented with topGO R packages based on the Kolmogorov–Smirnov test [48] and generated a directed acyclic graph. The Kyoto Encyclopedia of Genes and Genomes (KEGG) [49] is a database resource for understanding the high-level functions and utilities of a biological system. We used KOBAS software [50] to test the statistical enrichment of the differential expression genes in KEGG pathways. GO and KEGG terms with a corrected *p*-value below 0.05 were considered as significantly enriched for the DEGs.

### 4.6. Validation of RNA Sequencing Results Using qRT-PCR

The root samples of Cd-treated wheat at different growth days and the control were collected, and total RNA was extracted using the same protocol described above. To validate the expression of the candidate DEGs, eight genes were randomly selected to be analyzed by qRT-PCR. Forward and reverse primers for DEGs were designed using Primer Premier 5.0 (http://www.PremierBiosoft.com, accessed on 26 September 2019). The accuracy of the primers was also confirmed by using the NCBI Primer-BLAST tool. The reverse-transcription reactions were performed with 2 μg of RNA, using the iScriptTM Advanced cDNA Synthesis Kit (Hercules, CA, USA) with the Light Cycler 96 system (Roche). Each PCR reaction contained 20 µL of the total reaction volume, with 2 μL of diluted cDNA, 0.5 μL of forward and reverse primers, 10 μL of MASTER MIX (Thermo, Waltham, MA, USA) and 7 μL of dd H_2_O, with the following cycling conditions: 94 °C for 2 min, 35 cycles at 94 °C for 15 s, 60 °C for 15 s, and 72 °C for 20 s. The relative quantification of the target genes was calculated by the 2^−ΔΔCT^ procedure [51].

## 5. Conclusions

Our present study demonstrated 40 ul/L Cd stress-induced changes in the gene expression profiles of wheat seedlings over different seedling growth days. Different seedling days showed different intensities of the genes and pathways involved in the response to Cd stress. The data generated in our study can be a valuable resource and support genome analysis, besides aiding in the development of expression analysis platforms, identification of molecular marker development and initiating functional and comparative genomic studies. The DEGs involved in ABA hormones, metal ion transport, auxin, ABC transporters, HMA proteins and ROS play critical roles in resisting Cd stress. In addition, many new gene families have been identified in this study, but further experiments will be performed to investigate the mechanism of these DEGs in controlling Cd accumulation in wheat at the seedling stage. Selected DEGs were validated by qRT-PCR, which further contributed to strategies for improving the Cd tolerance in wheat.

## Figures and Tables

**Figure 1 plants-12-00642-f001:**
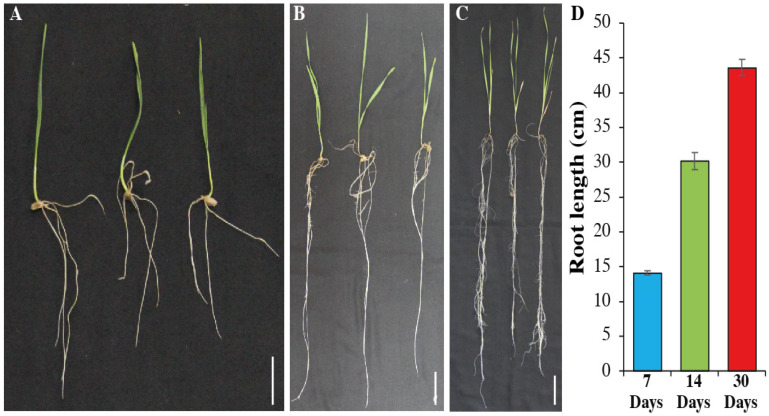
Root length of wheat seedlings at three different points. Seedlings at 7 days (**A**), 14 days (**B**), and 30 days (**C**). Bars = 4 cm. Quantification of root length (**D**). Bars are the mean of three replications and standard deviation was used.

**Figure 2 plants-12-00642-f002:**
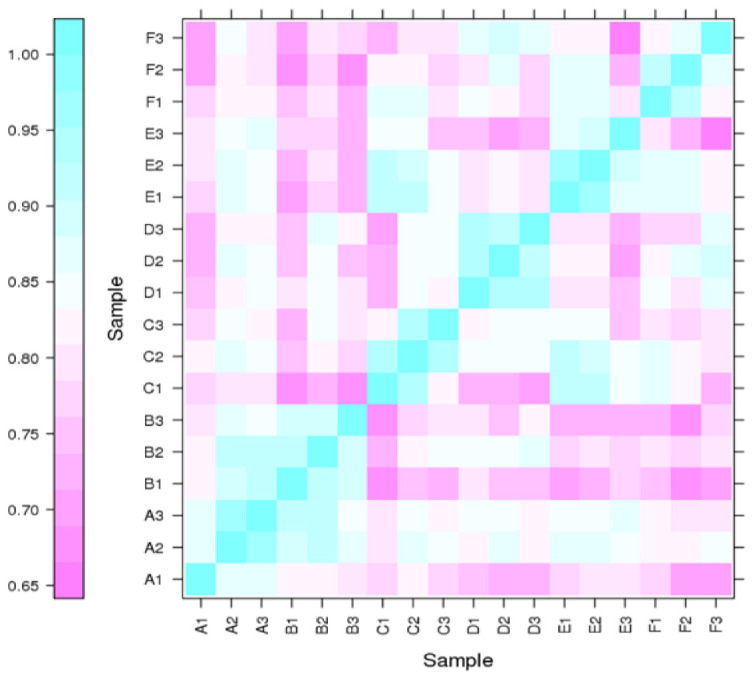
Correlation among RNA-Seq libraries at different time points of Cd stress. The *X*- and *Y*-axes represent each sample in a different time period. The expression values are log2-transformed median-centered FPKM. Cyan and purple color intensity indicates contig up-regulation and down-regulation, respectively.

**Figure 3 plants-12-00642-f003:**
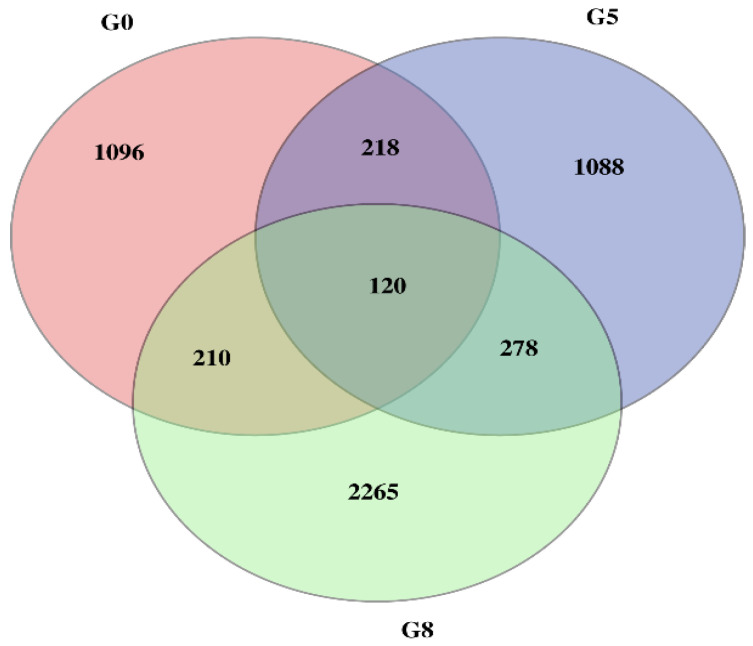
Venn diagram of differentially expressed genes from three comparative stages under Cd stress. Differential gene expression obtained by Venn diagram analysis. The number of differential gene expression profiles in the three libraries was obtained from the three comparative stages under Cd stress. The number of DEGs that overlap between G0 (7d), G5 (14d) and G8 (30d).

**Figure 4 plants-12-00642-f004:**
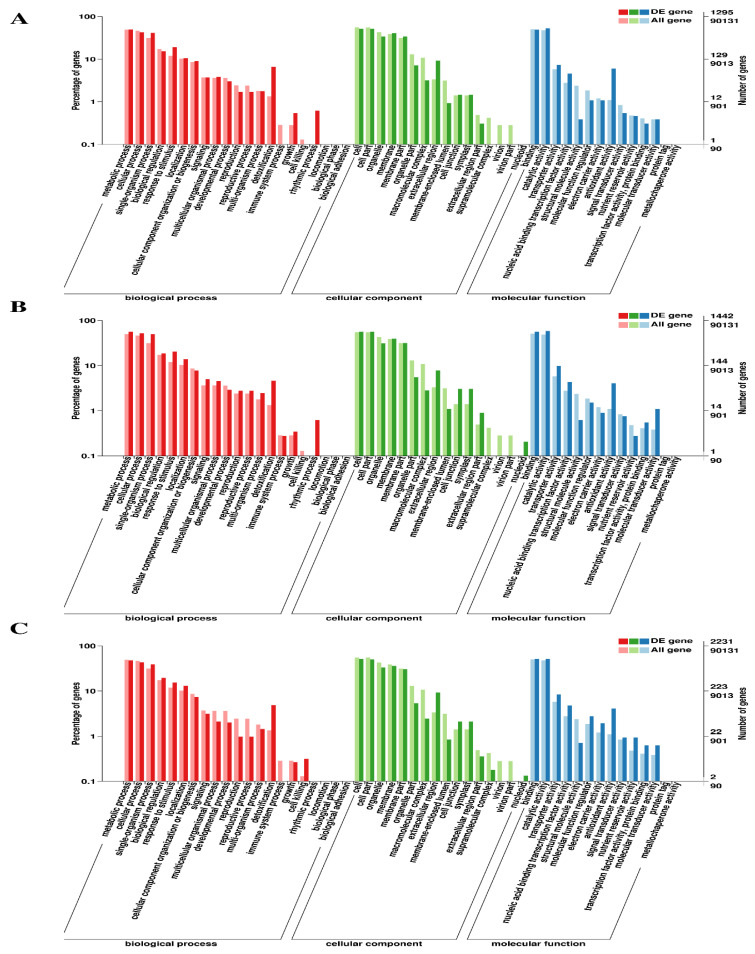
Functional classification of DEGs into different categories based on gene ontology (GO). The GO terms are classified into three main categories (biological process, cellular component and molecular function) and further into subcategories, which are indicated on the Y-axis. (**A**) At 7d of Cd stress; (**B**) at 14d of Cd stress; (**C**) at 30d of Cd stress.

**Figure 5 plants-12-00642-f005:**
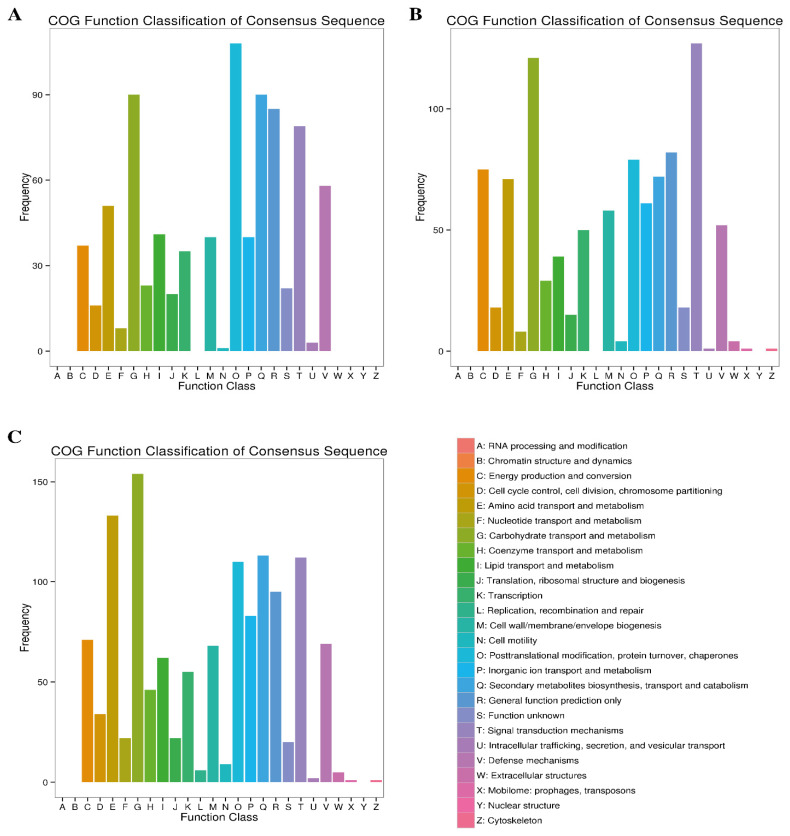
Clusters of orthologous gene (COG) annotation of DEGs at three comparative stages under Cd stress. (**A**) At 7d of Cd stress; (**B**) at 14d of Cd stress and (**C**) at 30d of Cd stress.

**Figure 6 plants-12-00642-f006:**
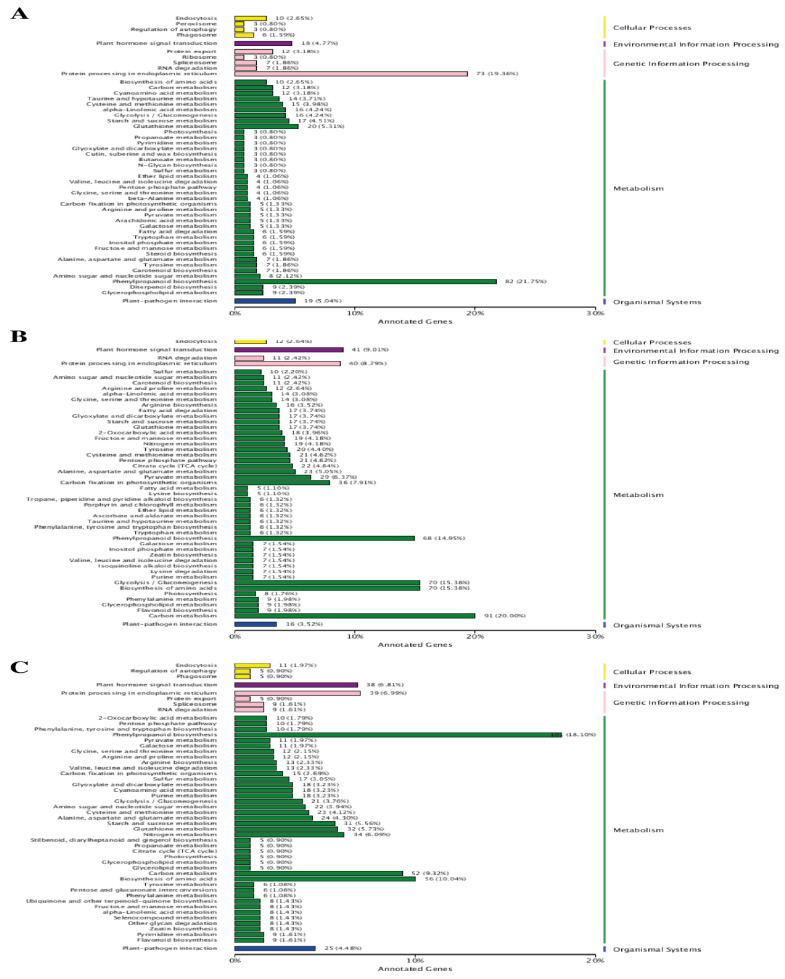
KEGG pathway annotation of DEGs at three comparative stages under Cd stress. (**A**) At 7d of Cd stress; (**B**) at 14d of Cd stress and (**C**) at 30d of Cd stress.

**Figure 7 plants-12-00642-f007:**
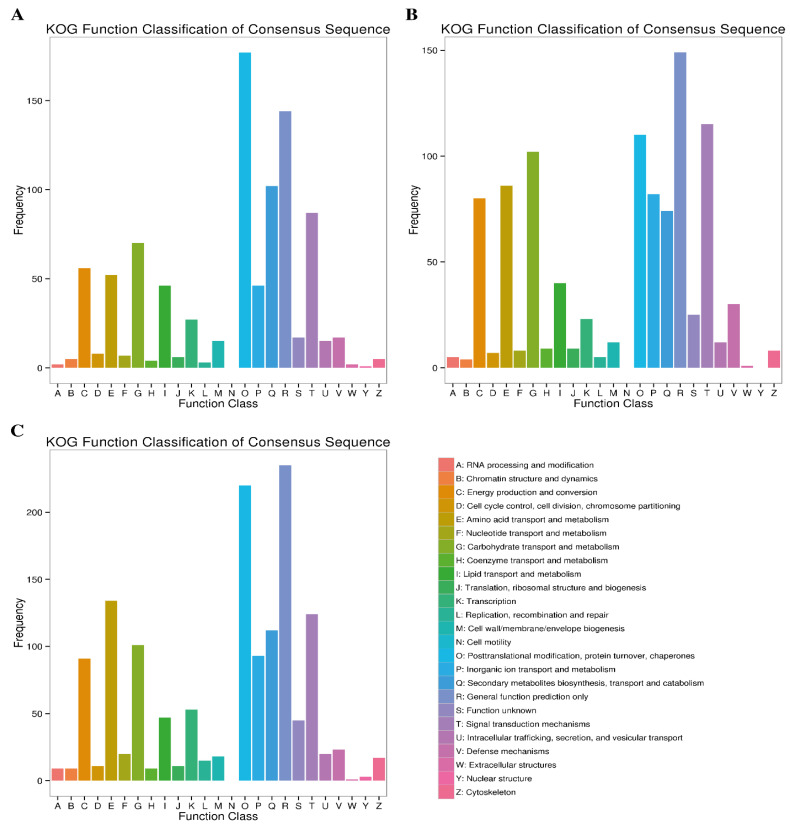
KOG pathway annotation of DEGs at three comparative stages under Cd stress. (**A**) At 7d of Cd stress; (**B**) at 14d of Cd stress and (**C**) at 30d of Cd stress.

**Figure 8 plants-12-00642-f008:**
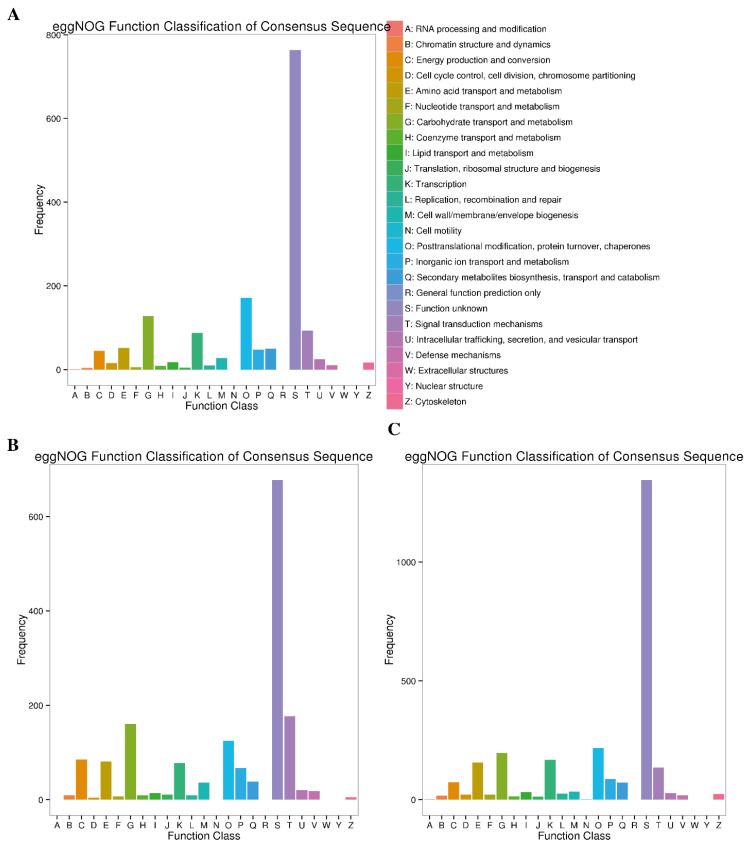
eggNOG pathway annotation of DEGs at three comparative stages under Cd stress. (**A**) at 7d of Cd stress; (**B**) at 14d of Cd stress and (**C**) at 30d of Cd stress.

**Figure 9 plants-12-00642-f009:**
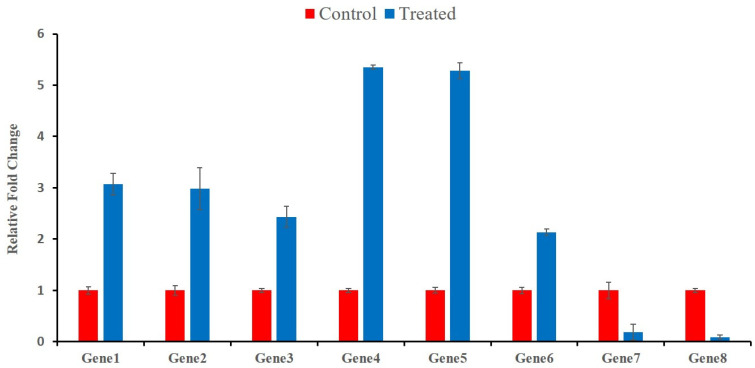
qRT-PCR validation results. The abscissa shows the genes’ ID and the ordinate shows the fold change. Quantification of gene expression was performed by the comparative 2^−ΔΔCT^ method. The error bars indicate the standard deviations of the three independent qRT-PCR biological replicates. Gene1 refers to TraesCS7A02G468500, Gene2 refers to TraesCS7A02G173900, Gene3 to TraesCS5A02G272800, Gene4 to TraesCS2A02G032200, Gene5 to TraesCS2A02G032100, Gene6 to TraesCS2D02G073900, Gene7 to TraesCS2B02G184400 and Gene8 to TraesCS5D02G004400.

**Table 1 plants-12-00642-t001:** Summary of total and clean reads from 18 comparative libraries.

Treatment	Total Raw Reads	Mapped Reads (%)	Unique Match (%)	Multiple Position Match (%)	% ≥Q30	GC Content (%)
A1_Cd stress	114,896,042	87.1	82.5	4.5	93.2	56.2
A2_Cd stress	112,982,852	89.1	85	4	92.9	55.7
A3_Cd stress	146,509,466	91.1	86.7	4.3	92.8	55.8
**Average**	124,796,120	89.1	84.7	4.3	93.0	55.9
B1_Normal	134,077,278	90.6	85.9	4.6	92.9	56.3
B2_Normal	130,033,680	85.2	80.5	4.6	93.1	56.1
B3_Normal	148,821,300	90.8	86.3	4.4	92.5	56.1
**Average**	137,644,086	88.9	84.2	4.5	92.8	56.2
C1_Cd stress	149,042,956	88.9	84.6	4.3	92.8	56.2
C2_Cd stress	148,930,020	88.1	83.3	4.7	93.1	56.2
C3_Cd stress	140,024,870	86.1	81.4	4.7	94	55.9
**Average**	145,999,282	87.7	83.1	4.6	93.3	56.1
D1_Normal	146,473,396	88.3	83.7	4.5	92.5	55.1
D2_Normal	146,020,820	86.8	82.7	4.1	93.9	54.5
D3_Normal	144,616,196	79.3	75.1	4.2	93	54.8
**Average**	145,703,471	84.8	80.5	4.3	93.1	54.8
E1_Cd stress	118,898,766	86.4	82.6	3.8	94	54.7
E2_Cd stress	133,663,600	87.6	84	3.7	93	56
E3_Cd stress	148,122,704	85.4	81.7	3.7	92.6	55.9
**Average**	133,561,690	86.5	82.8	3.7	93.2	55.5
F1_Normal	134,618,642	90.6	85.9	4.6	92.6	56.6
F2_Normal	139,259,934	88.3	84.3	3.9	93.2	54.6
F3_Normal	145,833,104	88.9	84.8	4	93.2	55
**Average**	139,903,893	89.3	85.0	4.2	93.0	55.4

A: Cd stress and B: normal at 7 days, C: Cd stress and D: normal at 14 days; E: Cd stress and F: normal at 30 days.

**Table 2 plants-12-00642-t002:** Summary of DEGs from three comparative time points under Cd stress.

DEG Set	DEG Number	Up-Regulated	Down-Regulated
Cd stress _vs_ Normal at 7d	1644	462	1182
Cd stress _vs_ Normal at 14d	1704	245	1459
Cd stress _vs_ Normal at 30d	2873	836	2037
**Average**	**2074**	**514**	**1559**

**Table 3 plants-12-00642-t003:** Summary of annotated DEGs at three different points.

Treatments	All	Up-reg.	Down-reg.	COG	GO	KEGG	KOG	pfam	SWISS	eggNOg	Nr
**A123 vs. B123**	1615	456	1159	714	1295	554	786	1421	1286	1286	1612
**%**	98.2	27.7	70.5	43.4	78.7	33.7	47.8	86.4	78.2	78.2	98
**C123 vs. D123**	1680	237	1443	848	1442	658	903	1514	1433	1615	1675
**%**	98.5	13.9	84.6	49.7	84.6	38.6	52.9	88.8	84.1	94.7	98
**E123 vs. F123**	2816	806	2010	1106	2231	891	1291	2389	2187	2641	2809
**%**	98	28	69.9	38.5	77.6	31	44.9	83.1	76.1	91.9	97

A: Cd stress and B: normal at 7 days, C: Cd stress and D: normal at 14 days; E: Cd stress and F: normal at 30 days.

## Data Availability

The transcriptome data is submitted to National Genomics Data Centre (https://ngdc.cncb.ac.cn/, Accessed on 12 August 2022) with Bio-project number PRJCA009646. Rest of the data presented in this study are within the main text as well as in the Appendix A.

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
