# Peer review of "Temporal Comparative Transcriptome Analysis on Wheat Response to Acute Cd Toxicity at the Seedling Stage"

_plants, 2023, doi:10.3390/plants12030642_

Round 1

Reviewer 1 Report

In this study, Imdad Ullah Zaid et al. reported the transcriptomic analysis of wheat seedlings in response to cadmium stimuli. 18 libraries from control and Cd stressed groups at different time points were generated for sequencing. Global analysis revealed that some of the hormone and metabolic biological pathways are changed in response to Cd stimuli. In general, the RNA-seq analysis performed here is too simple, stayed on the surface and there is much space to dig into. Though authors showed many GO enrichment and pathway changes, it seems like these changes didn’t led to a specific mechanism that can be rigorously tested. In the validation part, only several randomly selected genes in the changed pathways were used for qPCR, without further demonstrating the biological significance and function. Authors should take greater efforts to mining their data, generate specific hypothesis from the analysis, and provide convincing evidence that the identified molecule or pathway is truly contributing to the defective phenotypes of wheats under Cd stress. Therefore, I suggest rejection of the current manuscript.   

Author Response

Thank you very much for reviewing our paper. in this study we identifed several DEGs and their pathways through analysing different anotation platforms. yes we agree that in this paper we did not provide the very detail pathways of Cd stress related genes/candiadte genes of Cd tolerance in genes. but our study acute/nominate several genes pathways that were responsible in tolaering Cd stress in wheat. in the revised version we carefully adress the reviewers suggestions and points. crtical modifcation in main text was made. 

Reviewer 2 Report

The work applies transcriptomics for the analysis of differential expression in wheat seedlings exposed to Cd contamination. The paper is generally well presented, the reviewer has only a few questions/suggestions for the authors:

- In the abstract instead of just listing the number of differentially expressed DEGs in the various pathways one could refer specifically to a few genes of interest.

- "annotation" and "validation" are not satisfactory Keywords

- L77/80 it would be appropriate to indicate the scientific names of the plant species

-L105 add reference

-Table1: do the values indicated in rows 4-8-12-16-20-24 indicate the averages of the three previous values? please specify in the table

-Figures 3 and 5 improve text quality of histograms

- can you add in supplementary the root images obtained with image J?

-L464 superfluous statement

Consider implementing the bibliography by evaluating these papers:

Translated with www.DeepL.com/Translator (free version)Tartaglia, M., Sciarrillo, R., Zuzolo, D., Postiglione, A., Prigioniero, A., Scarano, P., Ruggieri, V., Guarino, C., 2022. Exploring an enhanced rhizospheric phenomenon for pluricontaminated soil remediation: Insights from tripartite metatranscriptome analyses. Journal of Hazardous Materials 428, 128246. https://doi.org/10.1016/j.jhazmat.2022.128246

Fan, W., Liu, C., Cao, B., Ma, S., Hu, J., Xiang, Z., Zhao, A., 2021. A meta-analysis of transcriptomic profiles reveals molecular pathways response to cadmium stress of Gramineae. Ecotoxicology and Environmental Safety 209, 111816. https://doi.org/10.1016/j.ecoenv.2020.111816

Yu, G., Ullah, H., Wang, X., Liu, J., Chen, B., Jiang, P., Lin, H., Sunahara, G.I., You, S., Zhang, X., Shahab, A., 2023. Integrated transcriptome and metabolome analysis reveals the mechanism of tolerance to manganese and cadmium toxicity in the Mn/Cd hyperaccumulator Celosia argentea Linn. Journal of Hazardous Materials 443, 130206. https://doi.org/10.1016/j.jhazmat.2022.130206

Wang, Y., Li, M., Liu, Z., Zhao, J., Chen, Y., 2021. Interactions between pyrene and heavy metals and their fates in a soil-maize (Zea mays L.) system: Perspectives from the root physiological functions and rhizosphere microbial community. Environmental Pollution 287, 117616. https://doi.org/10.1016/j.envpol.2021.117616

He, L., Li, J., 2022. Integrated transcriptome and physiological analysis of rice seedlings reveals different cadmium response mechanisms between indica and japonica varieties. Environmental and Experimental Botany 204, 105097. https://doi.org/10.1016/j.envexpbot.2022.105097

Author Response

Reply to reviewer 2
Point 1. In the abstract instead of just listing the number of differentially expressed DEGs in the various pathways one could refer specifically to a few genes of interest.
Answer. Thank you. We revised the abstract and remove the number of DEGs identified in our study. However, few numbers were still present in abstract that will provides actual number of identified DEGs in our study.
Point 2. "annotation" and "validation" are not satisfactory Keywords
Answer. Thank you. Both the keywords were removed and new key words are added
Point 3. L77/80 it would be appropriate to indicate the scientific names of the plant species and L105 add reference
Answer. Thank you. Both correction was made in the revised version
Point 4. Table1: do the values indicated in rows 4-8-12-16-20-24 indicate the averages of the three previous values? please specify in the table
Answer. Thank you. Following reviewer suggestion the table 1 was revised.
Point 5. Figures 3 and 5 improve text quality of histograms
Answer. Thank you. Quality and text of figures were improved by resizing it. However we merge three different figures and combine them to one.
Point 6. Can you add in supplementary the root images obtained with image J?
Answer. Thank you very much. This suggestion of reviewer was consider seriously and seedlings pictures along with root length data of three different points were added
Point 7. L464 superfluous statement
Answer Statement was revised

Reviewer 3 Report

The current manuscript is quite interesting in it approach to understand changes in Wheat plants in response to Cd at various stages of its growth. I have a few recommendations to further improve the manuscript and request the authors to consider them.

1. I understand that the Cd stress was imposed on separate sets of Wheat plants at 7 days of growth, 14 days of growth and 30 days of growth. Please confirm. If this is true there are several places where the reader can understand it as 7, 14 and 30 days of exposure to stress. Also there are many instances where 7d, 10d and 15d are used. Please re-check the confirm the data. Ex, at page 3 line 115-116; page 4 line 142 "7d, 10d and 15d"

2. May I please know the rationale behind measuring the root length and diameter in the present experimental plan? Also, please give data as MEAN±SD. The presented data suggests the measurements were made only for one seedling.

3. Table 1. Please explain the sample IDs in a footnote below the table. 

4. Table 2. Please add a footnote below the table explaining the DEG set.

5. Figure 1. Please try to highlight statistically significant correlation boxes and add values. This can improve the purpose of the figure.

6. Figure 2. Please explain what G0, G5 and G8 stand for. I am unable to find it in the manuscript.

7. Table 3. Same comment as 3 &4 

8. Figure 5. The clarity of the image could be improved as it is hard to read any text 

9. qPCR validation of RNA-Seq data is not required as I believe the same RNA sample used for RNA-Seq would have been used for cDNA synthesis for qPCR. For further reading please refer to 10.1016/j.bioflm.2021.100043

10. Please add the gene names if you would like to continue with presenting the qPCR data.  

All the best

Author Response

Reply to Reviewer 3
The current manuscript is quite interesting in it approach to understand changes in Wheat plants in response to Cd at various stages of its growth. I have a few recommendations to further improve the manuscript and request the authors to consider them.
1. I understand that the Cd stress was imposed on separate sets of Wheat plants at 7 days of growth, 14 days of growth and 30 days of growth. Please confirm. If this is true there are several places where the reader can understand it as 7, 14 and 30 days of exposure to stress. Also there are many instances where 7d, 10d and 15d are used. Please re-check the confirm the data. Ex, at page 3 line 115-116; page 4 line 142 "7d, 10d and 15d"
Answer. Thank you. We accept our mistake. In our study we used three points’ seedlings days (7d, 14d and 30d). Now all the corrections were made.
2. May I please know the rationale behind measuring the root length and diameter in the present experimental plan? Also, please give data as MEAN±SD. The presented data suggests the measurements were made only for one seedling.
Answer. Thank you. This suggestion of reviewer was consider seriously and seedlings pictures along with root length data of three different points were added.
3. Table 1. Please explain the sample IDs in a footnote below the table. 
Answer. Thank you. Table 1 is now revised and corrections were made
4. Table 2. Please add a footnote below the table explaining the DEG set.
Answer. Thank you. DEG set was revised and made cleared for readers
5. Figure 1. Please try to highlight statistically significant correlation boxes and add values. This can improve the purpose of the figure.
Answer. Thank you. In figure 2 (previous 1) the correlations were shown with color depth. Adding values at this stage will create confusion in figures.
6. Figure 2. Please explain what G0, G5 and G8 stand for. I am unable to find it in the manuscript.
Answer. Thank you. G0, G5 and G8 was explained in figure footnote.
7. Table 3. Same comment as 3 &4 
Answer. Thank you. Explained in figure footnote
8. Figure 5. The clarity of the image could be improved as it is hard to read any text 
Answer. Thank you. Quality and text of figure 6 was improved by resizing it. However we merge three different figures and combine them to one.
9. qPCR validation of RNA-Seq data is not required as I believe the same RNA sample used for RNA-Seq would have been used for cDNA synthesis for qPCR. For further reading please refer to 10.1016/j.bioflm.2021.100043
Answer. Thank you. We will like to carry on with Qpcr result. Therefor the gene names and relative information in the figure footnote was added.
10. Please add the gene names if you would like to continue with presenting the qPCR data.  
Answer. Thank you. qPCR data has significance in our study. In previous RNA seq studies qPCR analysis were used to validate the RNA seq data. Names of all eight genes were added in the footnote of figure 9.

Round 2

Reviewer 1 Report

Authors has addressed my previous essential concerns. This version is ready to go to the next step.

Author Response

thank you very much for your valuable comments and suggestion

Reviewer 3 Report

The current manuscript has been revised well. There are few recommendations yet to be addressed and request the authors to consider them to improve clarity

1. I understand that the Cd stress was imposed on separate sets of Wheat plants at 7 days of growth, 14 days of growth and 30 days of growth. Please confirm. If this is true there are several places where the reader can understand it as 7, 14 and 30 days of exposure to stress. Also there are many instances where 7d, 10d and 15d are used. Please re-check the confirm the data. Ex, at page 3 line 115-116; page 4 line 142 "7d, 10d and 15d"
Answer. Thank you. We accept our mistake. In our study we used three points’ seedlings days (7d, 14d and 30d). Now all the corrections were made.
Reviewer response: I observed. Thanks

2. May I please know the rationale behind measuring the root length and diameter in the present experimental plan? Also, please give data as MEAN±SD. The presented data suggests the measurements were made only for one seedling.
Answer. Thank you. This suggestion of reviewer was consider seriously and seedlings pictures along with root length data of three different points were added.
Reviewer response: I see that Figure 1 have been added. I understand there has been some misunderstanding. I see that there are at least 3 replications from the image. However, the parameter root length has only one value and no standard deviation data. I had raised this concern the last time as well. Please try to add the deviation data and scale bars for the images if possible. I think the images were not captured from the same distance (or height), so a scale bar would help if you want to keep the panels A,B and C in Figure 1.

3. Table 1. Please explain the sample IDs in a footnote below the table. 
Answer. Thank you. Table 1 is now revised and corrections were made
Reviewer response: I would recommend the authors to please add the foot note of Table 3 here as well. It will be easy to understand the data.

4. Table 2. Please add a footnote below the table explaining the DEG set.
Answer. Thank you. DEG set was revised and made cleared for readers

Reviewer response: Yes. Thank you.

5. Figure 1. Please try to highlight statistically significant correlation boxes and add values. This can improve the purpose of the figure.
Answer. Thank you. In figure 2 (previous 1) the correlations were shown with color depth. Adding values at this stage will create confusion in figures.
Reviewer response: I am sorry, I don’t think so. However, I had suggested it only to improve the quality of the figure and easily identify statistically significant correlations. It was only a recommendation and the authors can continue to use the same figure.  

6. Figure 2. Please explain what G0, G5 and G8 stand for. I am unable to find it in the manuscript.
Answer. Thank you. G0, G5 and G8 was explained in figure footnote.
Reviewer response: Yes. Thank you.

7. Table 3. Same comment as 3 &4 
Answer. Thank you. Explained in figure footnote

Reviewer response: Yes. Thank you.

8. Figure 5. The clarity of the image could be improved as it is hard to read any text 
Answer. Thank you. Quality and text of figure 6 was improved by resizing it. However we merge three different figures and combine them to one.

Reviewer response: Yes, I understand.

9. qPCR validation of RNA-Seq data is not required as I believe the same RNA sample used for RNA-Seq would have been used for cDNA synthesis for qPCR. For further reading please refer to 10.1016/j.bioflm.2021.100043
Answer. Thank you. We will like to carry on with Qpcr result. Therefor the gene names and relative information in the figure footnote was added.

Reviewer response: Yes. Thank you.

10. Please add the gene names if you would like to continue with presenting the qPCR data.  
Answer. Thank you. qPCR data has significance in our study. In previous RNA seq studies qPCR analysis were used to validate the RNA seq data. Names of all eight genes were added in the footnote of figure 9.

Reviewer response: Yes. Thank you.

Additional recommendations:

The Tables are not formatted in the same style. Please follow same style of formatting (borders at the least).

Author Response

thank you very much for your suggestions and corrections. In the revised version now the figure 1 was revised. The scale bar is now added in figure 1. Furthermore we will continue with the figure 2 as same as before.  the tables are now formatted in same styles.